# Factors affecting utilization of mental health services from Primary Health Care (PHC) facilities of western hilly district of Nepal

Gaurav Devkota[ID]*, Puspa Basnet, Bijay Thapa, Madhusudan Subedi

School of Public Health, Patan Academy of Health Sciences, Lalitpur, Nepal

* gdevkota74@gmail.com

## Abstract

### Aim

To explore the factors affecting mental health service utilization from Primary Health Care facilities of Arghakhanchi district, a western hilly district of Nepal.

### Background

Mental health service utilization has many facilitating and hindering factors present at different socio-ecological levels. Stigma and lack of awareness in the community have been identified as the major barriers for mental health service demand and access worldwide.

### Methods

A cross-sectional qualitative study was conducted in Arghakhanchi district of Nepal in July-August 2019 that collected information through face-to-face In-depth and Key Informant Interviews of three categories of participants selected judgmentally. Thirty-two purposively selected participants from the three categories were interviewed using validated interview guidelines. Thematic analysis was performed using RQDA package for EZR software. Validation of translated transcripts, member checking and inter-coder percent agreement were performed to maintain rigor in the study.

### Results

Mental health stigma and inadequate awareness were identified as major factors that caused barriers for mental health service utilization at community level. They also influenced different factors at other socio-ecological levels to act as barriers. Awareness in community along with accessibility and availability of comprehensive mental health services were recommended by the participants for increasing service utilization from Primary Health Care facilities.

**Data Availability Statement:** Data cannot be shared publicly as the information were audio-recorded and some of the participants may have willingly disclosed their identity. In addition, as the

manuscript is a part of the thesis conducted by corresponding author, all the information has been stored by the supervisor and Institutional Review Committee (IRC) of Patan Academy of Health Sciences (PAHS). Data are available from the Institutional Review Committee of PAHS (contact via irc-pahs@pahs.edu.np) as per the institutional rules for researchers who meet the criteria for access to confidential data. Patan Academy of Health Sciences is a public not-for-profit tertiary academic institution running School of Medicine, School of Public Health and School of Nursing. More information regarding Patan Academy of Health Sciences can be found from the following link: https://www.pahs.edu.np/ However, seven of the translated transcripts have been made available as supporting information.

**Funding:** GD was awarded "Dr. Harka Gurung- New ERA Fellowship 2019" for the research. New ERA is a non-government, non-profit research organization in Nepal (https://www.newera.com.np/). YES - Mr. Kiran Acharya of New ERA assisted in proof-reading of final manuscript that has been acknowledged in acknowledgement section of the original manuscript.

**Competing interests:** The authors have declared that no competing interests exist.

**Abbreviations:** AHW, Auxiliary Health Workers; ANM, Auxiliary Nurse Midwife; AUD, Alcohol Use Disorder; CHU, Community Health Unit; DALY, Disability Adjusted Life Years; DD, Depressive Disorder; EZR, Easy R; FCHV, Female Community Health Volunteer; HA, Health Assistant; HP, Health Post; ICR, Inter Coder Reliability; IDI, In-Depth Interview; IRC, Institutional Review Committee; KII, Key Informant Interview; LMICs, Low- and Middle-Income Countries; MNS, Mental, Neurological and Substance use; PAHS, Patan Academy of Health Sciences; PHC, Primary Health Care; PHCC, Primary Health Care Center; RQDA, R-based Qualitative Data Analysis; TB, Tuberculosis; UHC, Urban Health Clinic; YLD, Years Lost due to Disability.

## Conclusion

Individual, family and community awareness could help reduce and/or eliminate mental health stigma. Accessibility of health facilities and availability of comprehensive mental health services in Primary Health Care facilities could help increase service utilization from those facilities.

## Introduction

Inadequate mental health service utilization is a major factor among the many obstacles encountered in the control and management of mental illnesses. An estimated 300 million people are affected by depression, 60 million people by bipolar affective disorder, 23 million by schizophrenia and 50 million people by dementia worldwide [1]. National Mental Health Survey of Nepal, 2020 has identified prevalence of any mental disorder among adolescents aged 13 to 17 years as 5.2% (95% Confidence Interval: 4.2–6.4) and lifetime prevalence among adults as 10.0% (95% Confidence Interval: 8.5–11.8) and current prevalence as 4.3 (95% Confidence Interval: 3.5–5.2) [2, 3]. Mental, Neurological and Substance use (MNS) disorders is a term used by World Health Organization to incorporate mental, neurological, and substance use disorders in a single term which are often otherwise separated into treatment silos such as neurology, psychiatry, psychology, substance use, etc. in developed countries. Of all the people suffering from Mental, Neurological and Substance use (MNS) disorders, the proportion that had not received any treatment and care in past 12 month was of significant concern, being 35–50% in developed countries and 76–85% in less developed countries [4]. High treatment gap for mental illnesses has been identified from National Mental Health Survey of India (2015–2016) [5] and National Mental Health Survey of Nepal (2020) as well [3].

Stigma has been identified as a major barrier for mental health service demand worldwide [6]. Stigma can be defined as "a process involving labeling, separation, stereotype awareness, stereotype endorsement, prejudice and discrimination in a context in which social, economic or political power is exercised to the detriment of members of a social group" and can be of different types such as anticipated stigma, experienced stigma, internalized stigma, perceived stigma, stigma endorsement and treatment stigma [7]. Stigma and discrimination in Low- and Middle-Income Countries (LMICs) leading to pervasive human rights violations against people with mental and psychosocial disabilities act as barriers to mental health service usage [8].

Studies in Nepal have identified lack of awareness of mental health problem as major problem, mental health stigma and certain cultural norms as major barriers for accessing mental health services [9]. In addition, lacking financial means, fear of being perceived as weak, fear of being perceived as crazy and being too unwell to ask for help have also been identified as barriers to accessing mental health care [10]. Inadequate health literacy is also pivotal determinant for hindering access and engagement with health services as well as making and enacting health decisions [11]. Lack of availability of mental health services and senior staffs such as Senior Community Medicine Assistant, insufficient trained health workers, and mistreatment within health centers of rural areas have been identified as factors that influenced demand and access of mental health services at health facility level [9]. In addition, inaccessible mental health services, limited authority for prescription of psychotropic drugs, and knowledge and skill gaps of Primary Health Care (PHC) health workers on mental health have caused barriers in integration of mental health service as routine PHC care [12].

Strategies identified to increase the demand for mental health services include channeling mental health education through trusted and respected community figures, responding to the need for openness or privacy in educational programs, and adapting to local perceptions of stigmatized

treatments [9]. In addition, developing patient support group for collective organization and advocacy and a focal community resource person to aid in mental health service delivery and education; upgrading skills and knowledge of health workers through mhGAP resources (operations manual, training manual and intervention guide); and using mobile technology to deliver effective mental health services have been identified to improve mental health service utilization [12].

Non-reduction of treatment gap has implications beyond impact on national burden of diseases (Years Lived with Disability and Disability Adjusted Life Years), such as increasing indirect economic costs (owing to absenteeism and lost productivity at work) [13], impairing family function [14], and increasing the risk of teenage childbearing [15] and domestic violence [16]. Many socio-cultural and psychosocial barriers in demand for mental health care services as well as pragmatic and health system functioning barriers along with difficulties in accessing health care has widened the treatment gap of mental illnesses at PHC facilities. Therefore, in order to bridge treatment gap and shorten the treatment lag, it is essential to determine and pinpoint factors affecting mental health service utilization.

Many available works of literature on factors affecting mental health service utilization in Nepal focused on one or some of the socio-ecological levels. A qualitative formative study done to identify barriers and potential solutions for reaching people with priority mental illnesses identified barriers at community and health facility level [9], whereas another formative qualitative study exploring resources, challenges and potential barriers to develop primary care and community-based mental health services identified barriers at policy, health facility and community level [12]. Hence, this study was conducted in order to explore factors from all the socio-ecological levels that affect utilization of mental health services in Arghakhanchi district, a western hilly district of Nepal. The study was based on the socio-ecological model proposed by Urie Bronfenbrenner [17] that includes following five levels:

i.  individual level that includes characteristics that influences behavior such as knowledge, attitude and skill

ii.  interpersonal level that includes social networks like partners, friends and families

iii.  organizational level that includes formal and informal structures, environment and ethos

iv.  community level that includes established cultural norms and values

v.  policy level that includes national and local policies.

## Materials and methods

### Study design

The study was a cross-sectional exploratory qualitative study grounded in socio-ecological model that used an emic perspective for data collection and analysis. The study consisted of 16 In-Depth Interviews (IDIs) and 16 Key Informant Interviews (KIIs) conducted between July-August 2019. IDIs and KIIs were conducted in order to explore the knowledge, perception, and experiences of community on factors affecting utilization of mental health services from PHC facilities. In addition, recommendations for improving mental health service utilization from the PHC facilities in the district were explored.

### Operational definitions

In order to guide the selection of participants as well as to ease the conduction of the study, operational definitions were provided for the following terms before the start of information collection.

a. Primary Health Care Facilities: Public health care facilities, under local government and below District Hospital level, providing regular health care services. Primary Health Care Centers (PHCCs), Health Posts (HPs), Urban Health Clinics (UHCs) and Community Health Units (CHUs) were the PHC facilities visited.

b. People with mental illnesses: Patients under psychiatric medications for mental illnesses from any health facilities, whose identity were informed by Female Community Health Volunteers (FCHVs), PHC health workers or secondary health service providers.

c. Primary caretaker: Relatives of people with mental illnesses who have been continuously taking care of the patient from onset of the disease. Spouses in case of married, parents in case of children, in-laws, and siblings were interviewed.

d. Service providers: Health workforce who provided health services from the PHC facilities. Doctor, Nurse, Health Assistants (HA), Auxiliary Health Workers (AHW) and Auxiliary Nurse Midwife (ANM) were selected as they provided mental health services to the patients.

e. Health Administrators: Health coordinators of the municipalities and the head of Health Office in the district were termed as health administrators.

f. Elected Representatives: Mayors of municipalities and heads of rural municipalities who were elected for the positions after the local election held in Nepal in 2017.

## Setting

The study was conducted in Arghakhanchi district, a western hilly district of Nepal. The district not being selected in National Mental Health Survey as well as researchers' familiarity with the use of idioms and metaphors of the district influenced selection of the district as study site. Nearly 68% of the district lies in the mountainous Mahabharat Range and the rest lies in the Siwalik Hills. The district is surrounded by Palpa district in the east, Gulmi district in the north, Kapilvastu district in the south and Pyuthan district in the west. Population of Arghakhanchi district as per Census 2011 was 197,632 with 56% female, 49% under 20 years of age, life expectancy of 68.8 years, per capita income of 909 USD, 97% speaking Nepali, 51% of Brahmin-Chhetri, 28.8% poor, 73% able to read and write, and 2.6% with disability [18]. The major mental illnesses prevalent in the district are depression, psychosis, anxiety (neurosis), conversive disorder, alcoholism, epilepsy and mental retardation [19]. There were 46 Primary Health Care facilities in the district at the start of the study period [20]. Psychological counseling and referral services for people with mental illnesses were provided based on the Basic Health Service Package of Nepal, 2075 from those PHC facilities [21]. These services were provided regardless of the training and supervision the health care service providers had or did not have.

## Sampling procedure and sample size

For the purpose of the study, two rural municipalities (coded RM1 and RM2) among three rural municipalities and two municipalities (coded M1 and M2) among three municipalities within the district were selected conveniently as study sites (S1 Table). Judgmental sampling [22] was done to select three categories of participants (people with mental illnesses and/or their caretakers, PHC service providers and elected representatives/health administrators). Identity of people with mental illnesses was collected with the assistance of PHC health

workers, FCHVs, or secondary service providers. Accordingly, people with mental illnesses or their primary caretakers were recruited for IDIs purposively. Primary health care service providers and Health Administrators/Elected representatives were selected purposively. At the health facility level, two Health Posts and one Community Health Unit from RM1, one Health Post from RM2, one Health Post and one Urban Health Center from M1 and one PHCC from M2 were selected purposively in order to interview health care service providers. Health administrators and elected representatives of the conveniently selected rural municipalities and municipalities were interviewed. Thus, representation of both municipalities and all the level of PHC facilities were maintained.

Twelve primary caretakers and four people with mental illnesses, eight primary health care service providers and eight health administrators/elected representatives were interviewed and information saturation was maintained. Total number of interviews (32) was deemed to be sufficient for the objective of the study at the end of data collection. The study excluded eligible participants who did not provide consent, were unable to speak, were less than 18 years of age, or were seriously ill or with known severe illness.

## Data collection tools and processes

Socio-demographic characteristics of participants from all the categories were collected using an open-ended proforma. Interview Guide Approach [23] was used for face-to-face interviews that was audio-recorded. Face and content validity of the Interview Guidelines was maintained through repeated discussions with qualitative research experts, faculties of School of Public Health, psychiatrists of Patan Academy of Health Sciences and mental health research experts. Written consent was obtained from selected participants prior to information collection. Anonymity was maintained by providing codes to the interviewees and confidentiality was maintained by storing the interviews in a password-protected folder in a password-protected laptop. Ideas and knowledge obtained from the informal talks between researcher and community people as well as feelings and experiences the researcher had during the stay within the study district were recorded in a notebook. It helped to enhance the analysis process, which in turn, contributed to the quality of research findings.

For interviewing the health care service providers and health administrators/elected representatives, researcher went to their respective workplace in most of the cases. In some cases, health care service providers and health administrators/elected representatives were met in places where they felt comfortable to provide interview. Majority of interviews of caretakers were conducted at their homes. Some interviews of caretakers and all interviews of patients were conducted in the private clinic that they visited for treatment/follow-up. Interviews of people with mental illnesses were conducted in the clinic with psychiatrist being available in next room to assist during the interview, if required. However, all the interviews were conducted smoothly and did not require involvement of the psychiatrist.

## Ethical consideration

Approval for the study was obtained from the Institution Review Committee (IRC) of Patan Academy of Health Sciences (PAHS) [Ref: PHP1906141257.amend1]. Authorities of the selected municipalities in the district were well explained about the objectives of the study and written consent for conduction of the study was obtained. Informed consent form was designed in Nepali language and written consent was obtained from all participants prior to interview. The decision for participation in study was voluntary without any compensation for the time provided. Participants were informed about their right to withdraw from the study at any time without giving any justification. Collected audios and translated documents were

maintained with high privacy by the researcher. All the interviews of the participants were provided with interview codes during the translation of the interviews.

During the conduction of interviews with the primary caretakers of people with mental illnesses at their home or in the clinic, there was a tendency of the community people to be present during the interviews. In some cases, the researcher requested community people to provide privacy, which they obeyed. For rest of the cases and in case of interviews conducted in the clinic, the researcher took verbal consent from the interviewee when it was difficult to maintain privacy until a separate room was available. When a separate room was available, the interviews were conducted in that vary room.

Some of the participants who were selected for interviews were unable to read and write. In such cases, witnesses who could read and write were asked to read the consent in front of the participants and provide their signature on the consent form. Then, participants were asked for their consent and thumbprints were taken if they provided consent for the interview.

## Data analysis

The recorded interviews were directly translated into English as the researchers had same mother tongue in which the interviews were recorded. The process of direct translation provided time-efficiency. Another researcher, who was not involved in translation, performed verification of all the translated documents. The verified translated transcripts were imported to RQDA package [24] for EZR and were open coded manually after many readings and familiarization with the text. Thematic analysis with interpretivist approach [23] using socio-ecological lens was performed based on participants' knowledge, feelings, experiences and expectations as authentic source of information. Thematic analysis was based on the six-phase framework provided by Braun and Clarke moving forward and back between those phases [25]. Different codes that had connections were collected under predetermined broad themes, i.e., different levels of socio-ecological model.

Member checking has been identified to provide confirmability and credibility to the qualitative analysis [23]. In case of this study, member checking of the results was done by providing the results to some of the interviewed participants selected conveniently, who showed agreement on the results presented. In addition, two coders independently coded the interviews with the themes fixed *a priori* and no codes from the principal investigator were rejected. These codes were then validated by a qualitative research expert who also calculated the inter-coder reliability which showed an average positive percent agreement of 70.59%. Positive percent agreement was performed as there were only two coders and all the codes of Coder 2 matched with the codes of Coder 1 (Table 1).

## Results

Socio-demographics of the participants who provided interviews from categories health administrators/elected representatives, health care service providers and people with mental illnesses were collected whereas socio-demographics of the people with mental illnesses were collected when interviews were conducted with their primary caretakers. Socio-demographics of the participants have been depicted in Table 2 below.

Major findings of the study are presented in different pre-determined themes (Fig 1). The findings are followed by italicized texts which are the actual statements provided by the participants.

**Table 1. Inter-coder positive percent agreement.**

| S.N. | Themes | Coder 2 | Coder 1 | Percentage Agreement |
|------|--------|---------|---------|---------------------|
| 1. | Individual level factors affecting mental health service utilization | 7 | 11 | 63.64% |
| 2. | Interpersonal level factors affecting mental health service utilization | 4 | 6 | 66.67% |
| 3. | Organizational level factors affecting mental health service utilization | 4 | 7 | 57.14% |
| 4. | Community level factors affecting mental health service utilization | 6 | 7 | 85.71% |
| 5. | Policy level factors affecting mental health service utilization | 3 | 3 | 100% |
| | **Total** | **24** | **34** | **70.59%** |

## Factors affecting mental health service utilization

**Individual level.** Female sex, low economic status and lack of education were reported as factors hindering mental health service utilization.

*Additionally, if the patient is married female, then her family, relatives along with her husband could dominate the conditions of divorce or damage of family life could also happen. (Service Provider, 35 years, Female, Chhetri, Married)*

**Table 2. Socio-demographics of the participants.**

| S.N. | Socio-demographics* | Frequency |
|------|---------------------|-----------|
| **1** | **Age** | |
| | 0–14 years | 1 |
| | 15–59 years | 29 |
| | 60 years and above | 2 |
| **2** | **Sex** | |
| | Male | 17 |
| | Female | 15 |
| **3** | **Permanent address** | |
| | Rural Municipality | 14 |
| | Urban Municipality | 18 |
| **4** | **Ethnicity** | |
| | Brahmin | 18 |
| | Chhetri | 6 |
| | Janajati | 2 |
| | Dalit | 6 |
| **5** | **Religion** | |
| | Hindu | 31 |
| | Christian | 1 |
| **6** | **Marital status** | |
| | Unmarried | 3 |
| | Married | 26 |
| | Widowed | 3 |
| **7** | **Education status** | |
| | None | 6 |
| | Informal | 1 |
| | Formal | 25 |
| | **Total** | **32** |

* Socio-demographics recorded were of patients when the interviews were conducted with their caretakers

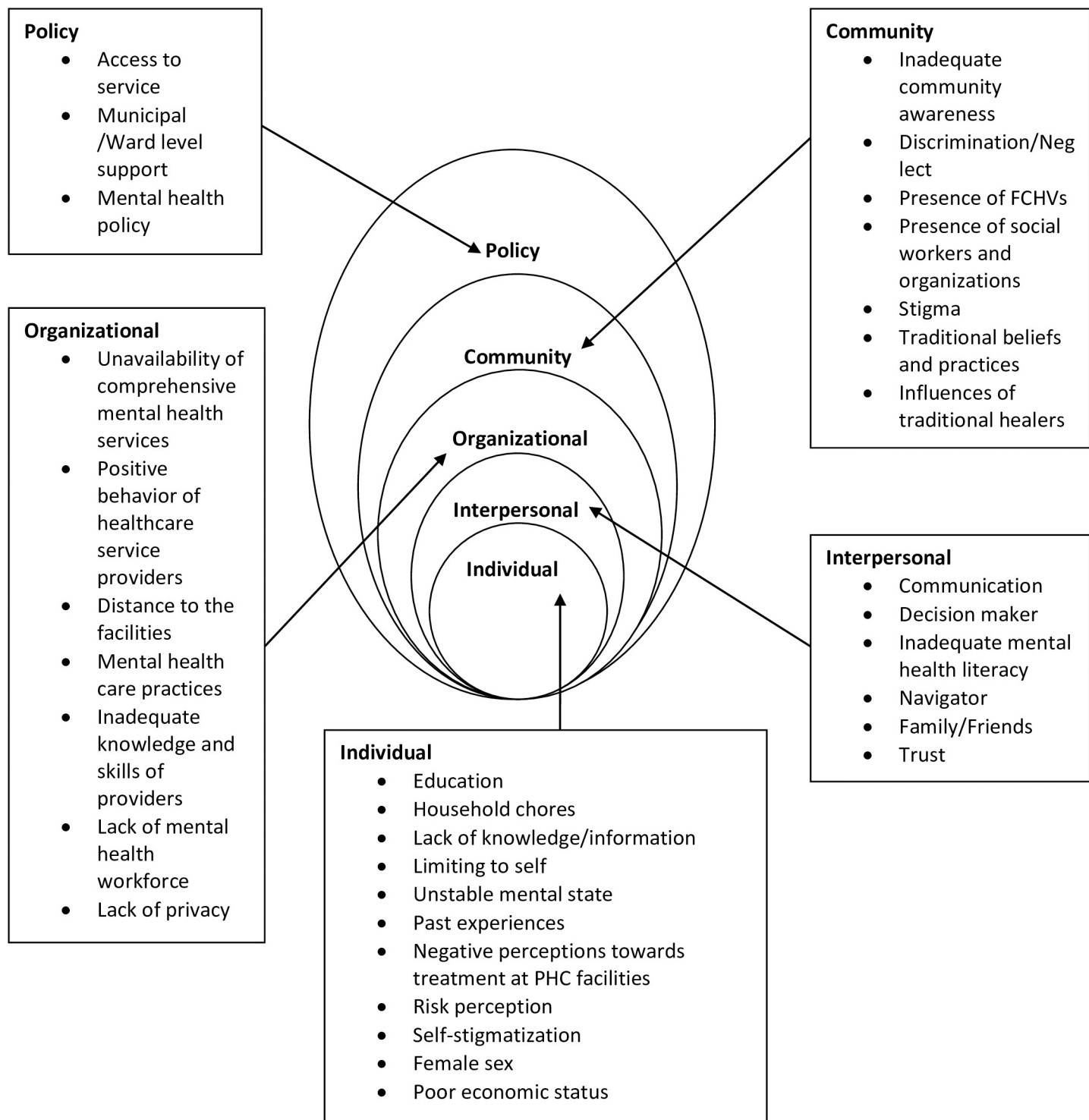

**Fig 1. Thematic analysis (factors affecting mental health service utilization).**

*If there are small health problems then they provide medicines from within the Health Post, but if there are some bigger health problems then they prescribe medicines from medicals saying there are no medicines in the Health Post for that disease. People having money can buy*

*those medicines but those who are poor cannot and have to die in vain. (Caretaker of 40 years old married Dalit female)*

*If the member is from an educated family, then they come themselves and ask about it otherwise they are not encouraged to take services. In contrast, they seek care from the "Dhami/ Jhakkri" rather than coming to us. (Service Provider, 25 years, Male, Brahmin, Unmarried)*

Household works also acted as barrier for mental health service utilization as caretakers had to provide time for taking care of children and/or livestock.

*I may not have visited PHC facilities because I am busy in my work. Even a married couple finds it hard to rear their child properly, I am alone, I should provide time to household works hence it is hard for me to provide time to take my sister to PHCC, HP or UHC. (Caretaker of 36 years old widowed Chhetri female)*

Inadequate knowledge and information among service users about availability of mental health services at PHC facilities acted as barrier. In addition, knowledge about availability of mental health specialists in different private clinics and hospitals also reduced service utilization from PHC facilities.

*We did not know whether mental health services were provided by PHCC or not. (Caretaker of 45 years old married Dalit female)*

Participants responded that not sharing of signs and symptoms by the patients and feeling shy to speak-up as barriers for mental health service utilization. However, people with mental illnesses who were able to identify their signs and symptoms and knew about the consequences of not treating the illness utilized mental health services.

*People with mental illnesses limit themselves to house and do not share their symptoms to anyone and thus they increase severity of the disease. (Health Administrator, 50 years, Male, Brahmin, Married)*

*Those who understand about the illness visit health facility thinking that they will be cured after treatments and some talk about symptoms of mental illness when they visit for check-up of other diseases. (Service Provider, 40 years, Female, Chhetri, Married)*

Ignorance and non-acceptance of the disease by people with mental illnesses due to fear of society and hiding the disease (self-stigmatization) were also identified as barriers for mental health service utilization.

*She used to refuse to go to the hospital saying she would be fine by herself. (Caretaker of 40 years old married Dalit female)*

*Next is ignorance from the patient self, they cannot accept themselves as having mental illness thinking what would people think of me, they will think that I am crazy, how would doctor treat me knowing my illness etc. (Health Administrator, 34 years, Female, Janajati, Married)*

*I do not know why we did not take him for treatment. . .. One thing is that he used to say he has no illness. . . (Caretaker of 34 years old married Brahmin male)*

Unstable mental state of the patients was also identified as barriers for service utilization because in such state it was either difficult to take them to health facilities or they would change their mind anytime.

*Mostly she used to reject to go to hospital for treatment. Sometimes she used to get ready to go and then suddenly used to change her decision. (Caretaker of 40 years old married Dalit female)*

People preferred health facilities where they had treated similar mental illnesses previously rather than visiting PHC facilities. Negative perception regarding treatment of mental illnesses available at PHC facilities also hindered mental health service utilization.

*As the symptoms were similar to that of my mother-in-law's we knew that it was mental illness and thus did not take my daughter to public health facilities of our rural municipality. (Caretaker of 3.5 years old Dalit female child)*

*It is because there are not good doctors at public health facilities and the medicines are not available there. After all public is public. (Patient, 41 years, Male, Chhetri, Married)*

*This clinic is nearer to us, also is Health Post, but at Health Post they provide opposite medicines, they only listen and provide medicines as per their wish. (Patient, 36 years, Female, Dalit, Married)*

**Interpersonal level.** Inability of family members and neighbors to identify mental illnesses at early stages acted as barrier for service utilization. Only after the disease is severe would they identify the disease and prefer mental health service utilization. People with more authority within house or in the neighborhood were identified as factors that would affect mental health service utilization from PHC facilities.

*The only thing that affected was we could not understand early that she was mentally ill. (Caretaker of 36 years old widowed Chhetri female)*

*. . .family members think that mentally ill person has no illness, is acting in order not to do works. . . (Service Provider, 40 years, Female, Chhetri, Married)*

*My husband used to say that there is no benefit taking her to Health Post. . . (Caretaker of 16 years old unmarried Dalit female)*

*To receive any services, not only mental health related, they first wait for their family decision and do what family decide to do at first. (Service Provider, 42 years, Female, Brahmin, Married)*

People had inadequate knowledge about different modalities of treatment for mental illnesses and thought medicines as the only means of treatment thereby not preferring PHC facilities. Therefore, inadequate mental health literacy in family members and neighbors was identified as one of the barriers for mental health service utilization.

*. . .there is belief in people that medicines must be prescribed when they go for check-ups and since medicines are not provided from primary health care facilities thus it has acted as barrier. (Health Administrator, 40 years, Male, Brahmin, Married)*

Communicating the outcome of treatment acted as both facilitator and barrier for service utilization. When the outcome was good, communication acted as facilitator and when the outcome was bad, it acted as barrier for mental health service utilization.

*My sister nowadays gives suggestions to other people about mental illness and its treatment. If she identifies any similarity as her symptoms in someone, she suggests meeting her doctor for treatment. (Caretaker of 36 years old widowed Chhetri female)*

Presence or absence of trust between mental health service provider and user were also identified as factor that affected mental health service utilization.

*I had done an insurance Sir, so they sent us to Dhorbas, Palpa. At Dhorbas there will be different doctor and the doctor will examine in their own way, and as the medicines she was taking were doing her good, I feared that the disease may return if the medicines are changed at Dhorbas, so I did not take her there. (Caretaker of 40 years old married Dalit female)*

*As I am from same local community, they (people with mental illnesses) easily share the problems they face. (Service Provider, 35 years, Female, Chhetri, Married)*

Navigators like teachers, neighbors and people in the community who provide ideas on places for treatment of mental illnesses were identified both as facilitator and as barrier for mental health service utilization from PHC facilities.

*Community leaders like teachers also help by providing suggestions for treatment in health facility. (Service Provider, 35 years, Male, Brahmin, Married)*

*There was no one suggesting that the treatment and cure of mental illness could be at Health Posts, so we took her to Bhairahawa as people said that cure was possible at Bhairahawa. (Caretaker of 38 years old widowed Brahmin female)*

*There are some relatives working in health posts who said that they do not provide medicines for mental illnesses from health post, so I never went there for treatment. (Patient, 66 years, Female, Brahmin, Widow)*

Support from family members, close acquaintances and neighbors by providing information, helping to reach health facility and access health services was identified as facilitator for mental health service utilization whereas neglect and lack of love from family and friends were identified as barriers for mental health service utilization.

*Support was from my brothers, no one else provided any support. Either her father or my brother has provided support; others have not provided support. (Caretaker of 16 years old unmarried Dalit female)*

*Many people in this community keep their family members suffering from mental illnesses at home tying and not bringing them to health facilities for treatment. (Service Provider, 35 years, Male, Brahmin, Married)*

*Hindrance was from my family as my in-laws were uneducated so they used to say not to take medicines, as it would make body stink and there were no one in the village to provide help. (Caretaker of 40 years old married Brahmin male)*

**Organizational level.**   Unavailability of comprehensive mental health services from the PHC facilities acted as barrier for mental health service utilization. However, counseling and suggestions by health care service providers regarding mental illnesses at health facilities or during fieldwork acted as facilitators for mental health service utilization.

*Public health facilities in this municipality do not provide medicines. There are no medicines for mental illness. (Caretaker of 50 years old married Janajati female)*

*If the medicines were provided at Health Post, then we need not come here [private clinic]. They do not provide medicines at Health Post so we come here. They provide 2 to 4 medicines for simple diseases but not of mental illness. (Patient, 66 years, Female, Brahmin, Widow)*

*Field visit by the health workers, counseling about their mental health problems during their visits at the health facility also motivate them and so they keep coming to take the health services in the municipality. (Health Administrator, 47 years, Male, Brahmin, Married)*

Commercialization of health care services, practice of prescribing psychotropic medicines in different private health facilities and clinics, and practice of referral of people with mental illnesses from PHC facilities to urban cities acted as barriers for utilization of mental health care services from PHC facilities. The presence of commercialization of mental health services in the district and referral to urban private health centers was also acknowledged by elected representatives and health administrator of the district.

*There is somewhat commercialization [of mental health services] in this rural municipality. (Elected Representative, 64 years, Male, Brahmin, Married)*

*We have also seen many women visiting Palpa for medication of mental illness, may be due to practice of prescribing medicines for mental illness there. (Health Administrator, 50 years, Male, Brahmin, Married)*

Location of PHC facilities also affected mental health service utilization. Participants responded that PHC facilities being far as barrier for service utilization. In addition, enrolling in health insurance made service users travel distant places in cases of referral thereby increasing their out-of-pocket expenses, and thus leading to lower service utilization from PHC facilities.

*We do not go to public health facilities as they are far from here. (Caretaker of 31 years old married Chhetri male)*

*. . .if anyone is enrolled in the insurance and is referred to other centers for treatment, only the cost for medicine is covered and not of transportation, for which we have to pay self, so it is not of help. (Caretaker of 40 years old married Dalit female)*

Inadequate mental health knowledge and counseling skills in mental health service providers at PHC facilities also acted as barriers for mental health service utilization. However, supportive and sympathetic nature of service providers facilitated mental health service utilization.

*But all the staffs over here are also not as capable and are not trained and aware about the problem yet, which may act as barrier. (Service Provider, 25 years, Male, Brahmin, Unmarried)*

*I took him to PHCC as well. Health care service providers over there provided good suggestions, also provided medicines if available, they used to support as much as possible, so it was for me to bring my husband to the condition where he is now. (Caretaker of 40 years old married Brahmin male)*

People's interest to seek mental health care services from experts and not from general health care service providers acted as barriers for mental health service utilization from PHC facilities. As some PHC facilities lacked separate rooms for diagnosing and counseling patients with mental illness, participants responded that lack of privacy while diagnosis and treatment also acted as barrier for mental health service utilization.

*There are no doctors for mental illnesses here at public health facilities, so I did not take my wife there for treatment. (Caretaker of 40 years old married Dalit female)*

*. . .to open up with the health worker for them there is no any privacy. . . (Service Provider, 39 years, Male, Brahmin, Married)*

## Community level

Stigma for mental illnesses was identified as the major factor that hindered mental health service utilization from PHC facilities and caused people to travel to distant places for mental health services.

*In the community, people who are educated take mental illnesses like any other illnesses but people in the community who are not educated think that those with mental illness have no sense and are crazy (pagal). (Caretaker of 45 years old married Brahmin female)*

*They say it is communicable, it is very dangerous, do not touch the patient, do not walk with them, do not have the meal on same plate or you may get the disease. (Caretaker of 3.5 years old Dalit female child)*

*. . .in our society, the perception towards the mental health patient is different from the perception towards normal person. (Health Administrator, 47 years, Male, Brahmin, Married)*

*. . .thinking that people will joke when they know about the illness within family, they do not visit primary health care facilities and if they have to, they go to distant health facilities for treatment and are under regular medicines without letting anyone in the community know about the illness. (Health Administrator, 50 years, Male, Brahmin, Married)*

Although mood disorders like depression was not stigmatized, psychosis was highly stigmatized, similar to diseases like TB and Leprosy.

*At community, there is trend of making minor illnesses as a severe one, like for TB, leprosy and for diseases like mental illnesses and dominate ill people. (Service Provider, 32 years, Male, Brahmin, Married)*

*There may be barriers like stigma for severe cases (psychosis) but for minor cases like depression there is no barriers, they come here. (Service Provider, 37 years, Male, Brahmin, Married)*

Inadequate awareness in community people regarding mental health and treatment of mental illnesses was identified as barrier for mental health service utilization from PHC facilities. Presence of discrimination in the community for people with mental illnesses and neglect

from the community regarding treatment of mental illnesses acted as barriers for mental health service utilization from PHC facilities.

> *Educated people and those who know about the disease take this illness as any other disease that can be cured with treatment and hence suggest to go for medications rather than "Lama-Jhakkri" but those who are not educated or have no knowledge about the disease talk about "risani", "laageko" and suggest for traditional healers. (Caretaker of 38 years old widowed Brahmin female)*

> *In community, after suffering from the mental health problem the perception towards him changes. And people even do not want to listen to him and says that his brain is weak, why to waste time by listening to what he is saying we have to go to our work. (Health Administrator, 47 years, Male, Brahmin, Married)*

Beliefs in traditional healers and practices of visiting traditional healers (*Dhami/Jhakri*) for treatment of mental illnesses in the community acted as barrier for mental health service utilization from PHC facilities. People also responded that taking to traditional healers did not provide any improvement of people with mental illnesses.

> *As my sister believed a lot in traditional healers it took us a long time to visit the hospital, it took us nearly 2 years. (Caretaker of 36 years old widowed Chhetri female)*

> *As we have customs in society to take such people to traditional healers thinking that the illness might be due to "bayu-batas", "bhoot-pichas", "risani-boksini" but there was no improvement seen. She got well by herself; taking to traditional healers did not help. (Caretaker of 42 years old married Dalit female)*

Presence of FCHVs in the community identifying people with mental illnesses, providing mental health awareness as well as suggestions to visit PHC facilities for treatment of such illnesses were identified as facilitators for mental health service utilization from PHC facilities. In addition, presence of social workers and different organizations within the district that help create awareness regarding mental health as well as provide suggestions to visit PHC facilities for treatment or coordinate with those facilities regarding treatment of people with mental illnesses acted as facilitator for mental health service utilization from PHC facilities.

> *Supporting factors are our FCHVs who are present in every toles, they recommend people with mental illnesses to visit health facilities to have check-ups and treatment from service providers at health facilities. . . .. there are many that support like Women development as well as there are many groups like health mother group, agricultural group. These groups have provided support. (Service Provider, 40 years, Female, Chhetri, Married)*

> *Social workers here are working for providing support to people with mental illnesses. If they find such person, they take them for the treatment at the service center or clinic. (Elected Representative, 58 years, Male, Brahmin, Married)*

> *Like this there are also group of female leaders who also used to refer cases by making communication with us. (Service Provider, 42 years, Female, Brahmin, Married)*

Participants identified that traditional healers have been acting both as facilitator and as barrier for mental health service utilization from PHC facilities. Some traditional healers suggested people with mental illnesses to visit health facilities in order to treat mental illnesses

whereas some traditional healers asserted that they could treat mental illnesses in shorter duration than health facilities and doctors.

> . . .patients first visit to Lama and he asks them to visit health facilities as well and they come here. (Service Provider, 35 years, Male, Brahmin, Married)

> I took her to a traditional healer (Buddha Lama) at Haraiya for "Jharfuk". He told me to keep my wife there for 7 days in order to cure her, and said that doctors would take 15 days to cure her. (Caretaker of 40 years old married Dalit female)

**Policy level.** Lack of health facilities within municipality as well as geographic constraints in access to health facilities were identified as barriers for mental health service utilization from PHC facilities.

> . . .the largest municipality of Nepal lies in this district; the municipality has such scattered community that it is hard for people there to utilize basic health services, even harder to get mental health services. (Health Administrator, 34 years, Female, Janajati, Married)

> Nepal's geography is still a constraint. There are many villages with lesser access of hospitals and many lack specialist doctors. . . (Caretaker of 45 years old married Brahmin female)

However, presence of roads and transportation facilities to different parts of the district were identified as facilitators for accessing mental health services.

> In spite of being Hilly district, there is development of road transport facility. (Service Provider, 35 years, Female, Chhetri, Married)

> There is availability of services to transport patients to district hospital or to nearby health posts. (Elected Representative, 64 years, Male, Brahmin, Married)

Financial support for mental health treatment from municipality/ward was identified as facilitator for mental health service utilization.

> If people with mental illnesses or their family members appeal for support from this rural municipality then we provide them with money for transportation and other support through a committee. (Elected Representative, 64 years, Male, Brahmin, Married)

> My sister said that municipality had provided some 5 to 10 thousand rupees as help after certifying some papers. (Caretaker of 45 years old married Dalit female)

However, there was also an instance when the support was not provided from the municipality/ward when asked for which may have caused barrier for mental health service utilization.

> Recently I talked with ward chief regarding some help for medicines and treatment but he said it was very difficult for recommendations at this time. . . (Caretaker of 38 years old widowed Brahmin female)

Participants reported that mental health services were provided as per National Mental Health Policy. However, as the policy for mental health had not been implemented properly,

**Table 3. Recommendations for improving mental health service utilization at PHC facilities.**

| Levels | Recommendations | Narratives |
|---|---|---|
| Individual | Awareness | *If people have awareness regarding disease condition, state of disease, knowledge about the proper treatment centers then we can empower them to visit health facilities for treatment. (Service Provider, 35 years, Male, Brahmin, Married)* |
| Interpersonal | Awareness | *For service users, if they or their family members could be assured that if they go to the health facility that their mental health problem will be solved then would they come to utilize the service. (Health Administrator, 47 years, Male, Brahmin, Married)* |
| Organizational | Trained mental health workforce, Infrastructures, Availability of medicines | *If doctors for mental illnesses were available at PHCC and Health Posts then it would be better for people of low economic status like us as we need not travel far for treatment and could get treatment easily. (Caretaker of 45 years old married Dalit female)*<br>*In addition, there should be availability of separate room for counseling as well as male health staff for treating male and female health staff for treating female patients with mental illnesses should be present. (Service Provider, 35 years, Male, Brahmin, Married)* |
| Community | Awareness | *I think that people with mental illnesses should not be tagged crazy (baulaha) or should not be neglected such that they die an unknown death or hurt any other people. (Caretaker of 36 years old widowed Chhetri female)* |
| Policy | Access to services | *Had there been a hospital at our municipality we would not have come here, we would spend our money within our village, and it would be a lot easier for us and would reduce our expenses on transportation. (Patient, 36 years, Female, Dalit, Married)* |

there existed barriers in mental health service delivery, thereby reducing service utilization from PHC settings.

> *There is no policy as facilitators and barriers of mental health services and not in process as well. We are working based on national policy. (Health Administrator, 40 years, Male, Brahmin, Married)*

> *As there is policy to deliver mental health services from PHC level, there is not much implementation of this policy to deliver mental health services. (Service Provider, 42 years, Female, Brahmin, Married)*

## Recommendations from the participants

Participants were also asked for recommendations in order to improve mental health service utilization from PHC facilities, thematic analysis of which were performed as per socio-ecological level and are depicted in Table 3.

## Discussion

This study identified predisposing factors like being female, having low economic status and lack of formal education as barriers for mental health service utilization, which shows partial similarity with the findings from a cross-sectional community survey done in Nepal [10] as the study identified that barriers to mental health care were not based on age, sex, marital status, education and caste/ethnicity but was associated with occupation. The differences present in the findings can be attributed to the fact that the previous study included only two disorders (Depressive Disorder and Alcohol Use Disorder) whereas the present study included different MNS disorders. However, a qualitative formative study done in Nepal at community and health facility level has identified that ignorance about mental health issues was highly present among people of high economic and caste status and among people with more education and resources [9]. Perception of disease (self-stigmatization) and limiting to self, perception regarding risk of disease, negative perception towards treatment of the disease, unstable mental state, and past experiences of treatment also acted as barriers for mental health service

utilization, findings that are similar to the study done in Nepal [10] and elsewhere [26]. Inadequate knowledge and information regarding mental health services as well as involvement in household chores created barriers for mental health service utilization from PHC facilities. Studies have shown that low demand of mental health care are present in people based at home, particularly females and those with no proper knowledge regarding correct place to seek help for mental illnesses [9].

Major finding from the study was mental health stigma in the community that was identified as the factor that itself acted as a barrier as well as influenced other factors at different socio-ecological levels. Self-stigmatization and limiting to self at individual level, lack of support from family and friends at interpersonal level, fear of privacy maintenance at organizational level, and discrimination and neglect at community level were influenced by stigma present in the community. Literature on underutilization of mental health services in India has concluded stigma as a combination of lack of knowledge (ignorance and misinformation), negative attitudes (prejudice) and excluding or avoiding behaviors (discriminations) [27]. Studies in different parts of world have also identified stigma as a major factor limiting mental health service utilization [6]. Inadequate awareness regarding mental health was identified as another major factor that acted as barrier for mental health service utilization as well as influenced traditional practices for treatment of mental illnesses. Majority of participants in this study recommended mental health awareness raising in order to reduce stigma thereby decreasing many barriers for mental health service utilization, which was also identified in different studies done in Nepal [9, 28].

Presence of FCHVs and social organizations and social workers in the community facilitated mental health service utilization whereas traditional healers played both supportive and hindering roles. Studies have identified that community informants are capable for accurate proactive case finding of mental disorders [29], and trained FCHVs are necessary condition to increase mental health service utilization as well as mental health service demand creation [28]. Use of trusted people in the community and respected public figures has been identified to cause largest impact on mental health awareness and thereby mental health service utilization [9, 30].

Support from family and friends were identified as factors facilitating mental health service utilization whereas lack of support, care, and love from the family hindered service utilization. Similar to these findings, a study has identified that family play dual role, on one hand they help to improve detection, access and maintenance of mental health treatment and care, and on the other hand ill-treat or neglect the patients due to fear of loss of status or being discriminated [30]. In addition, due to inadequate mental health literacy in community people, decision maker and navigators, suggestions visiting traditional healers or private clinics and hospitals were provided thereby causing barrier for mental health service utilization from PHC facilities. A rapid review done on the literatures that reported barriers to health care engagements in Nepal has recognized the importance of locally identified and developed health literacy in understanding, accessing and using health information and health services in Nepal [11]. In a formative study done in Nepal for setting mental health priorities, mental health literacy was identified as major strategy to overcome negative attitudes resulting in discrimination of people with mental health problems [30]. Communication of treatment successes from patients under treatment to patients who are not under treatment acted as facilitator for service utilization. It has been identified that mental health service users empathize with individuals experiencing mental health problems and help them understand mental health problems and ways to manage them [31]. Trust among service users and service providers facilitated mental health service utilization. A qualitative formative study done in Nepal has identified that misinformation about services, not knowing the correct place to seek help and

lack of trust in the quality and existence of services causes decrease in demand of mental health services from nearest PHC facilities [9].

Factors at organizational level hindering mental health service utilization were unavailability of comprehensive mental health services, lack of mental health specialists and psychotropic medicines, limited knowledge and skills of providers in mental health and provision of counseling and referral services only from the PHC facilities. Though Basic Health Service Package of Nepal 2075 has included basic mental health services to be provided from PHC facilities [21], and 16 psychotherapeutic medicines are listed in national essential medicine list of Nepal [32], participants responded lack of mental health services and psychotropic medicines at PHC facilities as factors hindering mental health service utilization. Similar findings were identified from different qualitative studies done in Nepal [12, 33]. PHC facilities being far and lack of proper infrastructure to maintain privacy also acted as barriers for mental health service utilization. Lack of accessibility and confidentiality and trust has been identified as barriers for mental health service utilization in previous studies [34]. However, sympathetic and supportive nature of the health care service providers facilitated utilization of mental health services available at PHC facilities. A study done in Nepal has identified that community people find it necessary that health workers were sensitive and maintained confidentiality in cases of mental health problems [30].

Inadequate implementation of the mental health policy at national and sub-national levels was the major factor at policy level that acted as barriers for mental health service utilization from PHC facilities. A qualitative study conducted in Nepal identified that availability of mental health policy and inclusion of mental health in other general policies and plans as a factor enabling good mental health system governance that may increase service utilization [33]. In addition, the geography and lack of health infrastructure within reach has been barriers for mental health service utilization from PHC facilities. However, some municipalities providing financial support or provision of transport have acted as facilitators for mental health service utilization. A study conducted to assess whether distance affects utilization of substance abuse and mental health services in the presence of transportation services among HIV-positive individuals did not identify distance as a barrier when the transportation services were provided [35]. Use of technology for the management of people with mental illnesses by psychiatrists without being physically present in the PHC facilities was also identified as factor that could facilitate mental health service utilization. Use of technology to provide mental health care has the potential to improve patient access to care and has been used effectively in rural areas, schools, forensic practices, and correctional facilities [36].

The study was conducted providing enough attention to its validity and reliability; however, there are strengths and limitations of the study. The study was a novel qualitative study that collected data from multiple sources to explore factors affecting mental health service utilization at PHC facilities of Arghakhanchi district. Data analysis performed was rigorous and was assisted by software. The study being a qualitative study with purposive sampling, findings cannot be generalized. However, the findings can be considered references to initiate studies and researches in similar contexts. Though the district has diversity in terms of socio-economy, culture and ethnicity, the study did not include representations from all the socio-demographics present in the district. The interviews were carried out in Nepali and directly translated to English and analyzed, which may have caused loss of original meaning of some rich and important information. Due to the nature of the study, subjectivity may not have been avoided during data analysis and interpretation, although measures (verification of translated transcripts, member checking of the results, inter-coder reliability, reflexivity and process documentation) were used to maintain rigor of the study.

## Conclusion

This study identified wide range of diverse but interweaving factors that affected mental health service utilization from PHC facilities. Stigma as a combination of lack of knowledge (ignorance and misconceptions), negative attitudes (prejudice) and excluding or avoiding behaviors (discriminations) is still present in the community acting as barrier for service utilization. Awareness (individual, family, and community) was a recurring term in the recommendations of the participants in order to reduce stigma and increase mental health service utilization. Accessibility of the health facilities and availability of comprehensive mental health services in PHC facilities could help increase mental health service utilization from the PHC facilities.

## Supporting information

**S1 Table. Primary Health Care facilities in Arghakhanchi district.**
(DOCX)

**S1 Appendix. Interview guidelines.**
(DOCX)

**S2 Appendix. Coding and codes within theme in EZR.**
(DOCX)

**S3 Appendix. Thematic mapping in EZR.**
(DOCX)

**S1 Transcript.**
(DOCX)

**S2 Transcript.**
(DOCX)

**S3 Transcript.**
(DOCX)

**S4 Transcript.**
(DOCX)

**S5 Transcript.**
(DOCX)

**S6 Transcript.**
(DOCX)

**S7 Transcript.**
(DOCX)

## Acknowledgments

We would like to acknowledge Dr. Rabi Shakya, Dr. Pawan Sharma, faculties of School of Public Health and Mr. Nagendra Prasad Luitel for their valuable inputs in face and content validation of the interview guidelines. We would like to thank Mr. Sudarshan Paudel for translation of the interview guidelines into Nepali and Ms. Reena Koju for back-translation. We also thank Dr. Phanindra Prasad Baral and Dr. Dipendra Gautam for providing useful insights and materials on national mental health policy and plans. The assistance provided by Dr. Amit Arjyal and Mr. Kiran Acharya for the preparation of final manuscript is highly appreciated.

## Author Contributions

**Conceptualization:** Gaurav Devkota, Bijay Thapa, Madhusudan Subedi.

**Data curation:** Gaurav Devkota, Puspa Basnet.

**Formal analysis:** Gaurav Devkota, Puspa Basnet.

**Funding acquisition:** Gaurav Devkota.

**Investigation:** Gaurav Devkota.

**Methodology:** Gaurav Devkota, Bijay Thapa, Madhusudan Subedi.

**Project administration:** Gaurav Devkota.

**Resources:** Gaurav Devkota.

**Software:** Gaurav Devkota, Puspa Basnet.

**Supervision:** Bijay Thapa, Madhusudan Subedi.

**Validation:** Gaurav Devkota, Puspa Basnet.

**Visualization:** Gaurav Devkota, Bijay Thapa, Madhusudan Subedi.

**Writing – original draft:** Gaurav Devkota.

**Writing – review & editing:** Gaurav Devkota, Puspa Basnet, Bijay Thapa, Madhusudan Subedi.

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
