## [Decision Letter · Decision Letter 0]

12 Jun 2020

PONE-D-20-06276

Factors affecting utilization of mental health services from Primary Health Care (PHC) facilities of western hilly district of Nepal: The roles of stigma, awareness and availability

PLOS ONE

Dear Dr. Devkota,

Thank you for submitting your manuscript to PLOS ONE. After careful consideration, we feel that it has merit but does not fully meet PLOS ONE’s publication criteria as it currently stands. Therefore, we invite you to submit a revised version of the manuscript that addresses the points raised during the review process.

We look forward to receiving your revised manuscript.

Kind regards,

Heidi H Ewen, Ph.D.

Academic Editor

PLOS ONE

Journal Requirements:

2. Please describe in your methods section how capacity to consent was determined for the participants in this study.

Reviewers' comments:

Reviewer's Responses to Questions

**Comments to the Author**

1. Is the manuscript technically sound, and do the data support the conclusions?

Reviewer #1: Yes

2. Has the statistical analysis been performed appropriately and rigorously? 

Reviewer #1: Yes

3. Have the authors made all data underlying the findings in their manuscript fully available?

Reviewer #1: Yes

4. Is the manuscript presented in an intelligible fashion and written in standard English?

Reviewer #1: Yes

5. Review Comments to the Author

Reviewer #1: Thank you for the opportunity to review this article focused on factors influencing mental health services utilization.

Background:

Importance of conducting research in this district not clear? Why rural vs urban area was chosen?

Explain what MNS mean and what disorders it represent. Dementia statistic was mentioned. Were people with dementia interviewed as well?

Please make sure you are consistent on wording either use mental illnesses or mental health disorders. If you are talking about mental illnesses in general term, I would suggest using it as plural term than singular.

Pg. 3 line 59, mental disorders should be mental health disorder (throughout the paper, this should be changed), but I would recommend either using mental illnesses throughout or mental health disorders throughout.

Pg. 4 Line 75: should be mental health care instead of mental care

Please provide some background information on mental health care provider in Nepal, in particular the district (region), where this study was conducted. Also, types of mental health disorders.

Pg 5, line 88, not sure what mhGAP resources means

Line 91: YLDs and DALYs, what do these terms mean?

Line 93: How is teenage childbearing related to treatment gap?

Please rewrite sentence starting at line 94 With difficulties even in….to make simple and clearer

Line 107: Please briefly describe Bronfenbrenner model.

I would refrain from using mental ill patients, please change these phrases (through out the paper) to patient with mental illness or patient with mental health disorder.

Methods:

Pg. 6 Operational definition, b. How did FCHVs identify patients with mental illnesses? Was this volunteer doing this for the research?

Pg. 7, line 143: I would recommend changing to 2.6% individuals with disability rather than disabled population.

Line 144-146: Does it mean that the district is not included in ongoing national mental health survey, is that one of the reasons to select this as a research site? Why is this district not included in national survey? This should be described as part of background and significance section. Also, please provide the statistics on MNS disorders in this district and region and how they differ from national data. This information should be in background.

Please explain if the participants with mental illnesses were already patients at the PHC. How were they identified/diagnosed?

Please make sure all the abbreviations are spelled out in their first use in the paper.

Table 1: this table can be used in supplement materials. I would rather have descriptive table in the manuscript rather than supplement. For demographic descriptive table in supplement (this should be moved to main document), I would use descriptive statistic in mean (SD), frequency (%), and/or range, rather than describing each participant. Provide descriptive statistic for each categories: mean age of participants, frequency for gender, ethnicity, religion, marital status, education, and occupation. I would provide frequency for different mental health disorders of the patients as well. It is not explained anywhere what is the mental health disorders of the participants interviewed.

Conceptual model should be part of the main paper as well, not supplement.

I had hard time figuring out thematic mapping and coding document. It would certainly help if the words and figure were clearer. However, I am not a qualitative researcher, so I would rather not comment on those two documents.

Were participants compensated for their time in anyway?

What does nearest caretaker mean? I think it could be substituted by primary caretaker? Is this person responsible for majority of the patient’s informal care? then I would substitute the nearest word to primary.

For quotes in italics, they should be indented as well.

Results:

Pg. 12, line 237, Did you mean “misunderstood mental illnesses as communicable disease”? or “treated mental illnesses as they were communicable diseases”?

If you are taking about mental illness as general term, I would write it as plural term: mental illnesses.

You should explain what “dhami/jhakri” mean for Non-Nepali readers.

Discussion:

What were the difference between the study you cited? It is mentioned that there was partial similarity, what was not similar? Again, what is MNS and what disorders it encompasses should be explained in background section.

Line 483-484: “highly present among people of high economic and caste status as well as those with more education and resources”: do you mean with both high and low economic and education resources?

6. PLOS authors have the option to publish the peer review history of their article (what does this mean?). If published, this will include your full peer review and any attached files.

Reviewer #1: No

---

## [Author Response · Author response to Decision Letter 0]

17 Oct 2020

We would like to thank the reviewer for reviewing our article titled “Factors affecting utilization of mental health services from Primary Health Care (PHC) facilities of western hilly district of Nepal: The roles of stigma, awareness and availability”. All the comments made are heartily acknowledged. Please find the responses on the comments made on your review. The marked-up copy and an unmarked version of revised paper has also been submitted for your perusal.

Background:

Comment:Importance of conducting research in this district not clear? Why rural vs urban area was chosen?

Response:The comments are well acknowledged and the necessary changes are made.

The importance of conducting the research is provided from Line 105 to 124which explains that only organizational, community and policy level factors from the socio-ecological levels had been identified in previous studies, thus this study was done to explore factors at all the levels of socio-ecological model proposed by Urie Bronfenbrenner (individual, interpersonal, organizational, community and policy level). 

The rationale for conducting the research in the district are provided under “Introduction” section of the manuscript, as suggested (line 112 to 115).

The study does not compare between rural and urban areas, however participants from both the areas were selected so as to provide representation from both the areas. This is provided in detail under “Sampling procedure and sample size” topic in “Materials and Methods” section (line 165 to 179).

Comment:Explain what MNS mean and what disorders it represents. Dementia statistic was mentioned. Were people with dementia interviewed as well?

Response: The comment is well acknowledged and the explanation has been provided.

MNS disorder was termed by World Health Organization in order to incorporate mental, neurological, and substance use disorders in a single term which are often otherwise separated into treatment silos such as neurology, psychiatry, psychology, substance use, etc. in developed countries.

Reference: Altevogt BM., Hanson SL., Ssali ZN, Cuff P, Rapporteurs. Mental, Neurological, and Substance Use Disorders in Sub-Saharan Africa: Reducing the treatment gap, Improving quality of care. 2001. ISBN: 9780309148801

People with dementia were not interviewed in the study. The statistics were provided as the five most common MNS disorders include dementia. People with mental illnesses who were recovering from their illnesses were only interviewed, in presence of psychiatrist in nearby room.

Comment:Please make sure you are consistent on wording either use mental illnesses or mental health disorders. If you are talking about mental illnesses in general term, I would suggest using it as plural term than singular.

Response: The comment is well acknowledged and the necessary changes have been made. The term “mental illnesses” has been used throughout the paper, as suggested.

Comment:Pg. 3 line 59, mental disorders should be mental health disorder (throughout the paper, this should be changed), but I would recommend either using mental illnesses throughout or mental health disorders throughout.

Response: The comment is well acknowledged and the necessary changes have been made. The term “mental illnesses” has been used throughout the paper, as suggested.

Comment:Pg. 4 Line 75: should be mental health care instead of mental care

Response: The comment is well acknowledged and the necessary change has been made, as suggested.

Comment:Please provide some background information on mental health care provider in Nepal, in particular the district (region), where this study was conducted. Also, types of mental health disorders.

Response: The comment is well acknowledged and the necessary information has been provided.

The type of mental illnesses within the district have been provided in the “Introduction” section, as suggested, along with the rationale for selection of the district for study.

The health care service providers have been listed in the “Service provider” sub-topic of “Operational Definition” topic in “Materials and Methods” section (Line 147 to 149).

Comment:Pg 5, line 88, not sure what mhGAP resources means

Response: The comment is well acknowledged and the necessary information has been provided.

mhGAP operations manual, training manual and intervention guide endorsed by World Health Organization are the mhGAP resources that are available to help reduce the mental health treatment gap and to enhance the capacity of Member states to respond to the large burden of Mental, Neurological and Substance use disorders. 

Reference: https://www.who.int/mental_health/mhgap/en/

Comment:Line 91: YLDs and DALYs, what do these terms mean?

Response: The comment is well acknowledged and the necessary changes has been made.

The full forms for the abbreviations have been provided in their first use.

YLD means Years Lived with Disability and DALY means Disability Adjusted Life Years.

Comment:Line 93: How is teenage childbearing related to treatment gap?

Response:The comment is well acknowledged. 

The statement was made based on a previous study.

Reference: Kohn R, Saxena S, Levav I, Saraceno B. The treatment gap in mental health care. Bull World Health Organ. 2004;82(11):858–66.

Comment:Please rewrite sentence starting at line 94 With difficulties even in….to make simple andclearer

Response: The comment is well acknowledged and the statement has been re-written as below.

Many socio-cultural and psychosocial barriers in demand for mental health care services as well as pragmatic and health system functioning barriers along with difficulties in accessing health care has widened the treatment gap of mental illnesses at PHC facilities.

Comment:Line 107: Please briefly describe Bronfenbrenner model.

Response: The comment is well acknowledged and the description is provided as below.

The five socio-ecological levels of the model in which the study is based upon (Bronfenbrenner model) are: 

i. individual level that includes characteristics that influences behavior such as knowledge, attitude and skill

ii. interpersonal level that includes social networks like partners, friends and families

iii. organizational level that includes formal and informal structures, environment and ethos

iv. community level that includes established cultural norms and values

v. policy level that includes national and local policies.

Comment:I would refrain from using mental ill patients, please change these phrases (through out the paper) to patient with mental illness or patient with mental health disorder.

Response: The comment is acknowledged and the necessary changes have been made throughout the paper, as suggested.

Methods:

Comment:Pg. 6 Operational definition, b. How did FCHVs identify patients with mental illnesses? Was this volunteer doing this for the research?

Response: The comment is well acknowledged and the following changes are made in the operational definition.

People with mental illnesses: Patients under psychiatric medications for mental illnesses whose identity were informed by Female Community Health Volunteers (FCHVs), PHC health workers or secondary health service providers.

Comment:Pg. 7, line 143: I would recommend changing to 2.6% individuals with disability rather than disabled population.

Response: The comment is well acknowledged and the change have been made as per suggestion.

Comment:Line 144-146: Does it mean that the district is not included in ongoing national mental health survey, is that one of the reasons to select this as a research site? Why is this district not included in national survey? This should be described as part of background and significance section. Also, please provide the statistics on MNS disorders in this district and region and how they differ from national data. This information should be in background.

Response: The comment is well acknowledged and the rationale for selection of the district as study site and the reason for the district not being included in the national survey has been provided in “Introduction” section as suggested.

Respected Reviewer, we would like to notify that Nepal does not have a national data on MNS disorders as well as statistics for the district, and thus the data could not be provided. The reason for the national mental health survey is itself to have a national statistic on mental health.

Reference: http://nhrc.gov.np/wp-content/uploads/2019/04/Pilot-national-mental-health.pdf

Comment:Please explain if the participants with mental illnesses were already patients at the PHC. How were they identified/diagnosed?

Response: The comment is well acknowledged. 

The operational definition for “People with mental illnesses” is provided in line 140-143 as

Patients under psychiatric medications for mental illnesses from any health facilities whose identity were informed by Female Community Health Volunteers (FCHVs), PHC health workers or secondary health service providers.

The operational definition is to clarify that these people were patients and under treatment at any health facilities. The information regarding people under psychiatric medications were provided by Female Community Health Volunteers (FCHVs), PHC health workers or secondary health service providers.

Comment:Please make sure all the abbreviations are spelled out in their first use in the paper.

Response: The comment is well acknowledged and the necessary changes are made as per suggestion.

Comment:Table 1: this table can be used in supplement materials. I would rather have descriptive table in the manuscript rather than supplement. For demographic descriptive table in supplement (this should be moved to main document), I would use descriptive statistic in mean (SD), frequency (%), and/or range, rather than describing each participant. Provide descriptive statistic for each categories: mean age of participants, frequency for gender, ethnicity, religion, marital status, education, and occupation. I would provide frequency for different mental health disorders of the patients as well. It is not explained anywhere what is the mental health disorders of the participants interviewed.

Response: The comments are well acknowledged and the necessary changes have been made.

The descriptive statistics has not been shown in mean (SD), frequency (%) and range owing to the purposive nature of participant selection. Moreover, the socio-demographics were gathered in order to provide background characteristics of “WHO” is providing the narration rather than for any comparisons.

In addition, the interviews were mostly conducted with primary caretaker of the people with mental illnesses and thus the exact type of the mental illnesses were not collected.

Comment:Conceptual model should be part of the main paper as well, not supplement.

Response: The comments are well acknowledged and the necessary changes have been made. The result of the thematic analysis based on the theoretical model has been provided as Figure 1 in the revised paper.

Comment:I had hard time figuring out thematic mapping and coding document. It would certainly help if the words and figure were clearer. However, I am not a qualitative researcher, so I would rather not comment on those two documents.

Response: The comment is well acknowledged and the changes have been made.

Respected Reviewer, the result from the thematic analysis has been provided as Figure 1 in the revised paper.

Comment:Were participants compensated for their time in anyway?

Response: The comment is well acknowledged. As the participation was completely voluntary thus no compensation was provided for the time provided by the participants.

Modification in the statement (line 212-213) has been made.

“The decision for participation in study was voluntary without any compensation for the time provided.”

Comment:What does nearest caretaker mean? I think it could be substituted by primary caretaker? Is this person responsible for majority of the patient’s informal care? then I would substitute the nearest word to primary.

Response: The comment is well acknowledged and the necessary changes has been made, as suggested.

Comment:For quotes in italics, they should be indented as well.

Response: The comment is acknowledged and the necessary changes have been made throughout the paper, as suggested.

Results:

Comment:Pg. 12, line 237, Did you mean “misunderstood mental illnesses as communicable disease”? or “treated mental illnesses as they were communicable diseases”?

Response: The comment is acknowledged and the change has been made to “misunderstood mental illnesses as communicable diseases”.

Comment:If you are taking about mental illness as general term, I would write it as plural term: mental illnesses.

Response: The comment is acknowledged and the necessary changes are made throughout the paper, as suggested.

Comment:You should explain what “dhami/jhakri” mean for Non-Nepali readers.

Response: The comment is well acknowledged and the traditional/faith healers are identified as “dhami/jhakri” in line 301-302.

Discussion:

Comment:What were the difference between the study you cited? It is mentioned that there was partial similarity, what was not similar? Again, what is MNS and what disorders it encompasses should be explained in background section.

Response:The comment is well acknowledged. The factors identified in the previous study has also been listed and changes made in the sentence.

Comment:Line 483-484: “highly present among people of high economic and caste status as well as those with more education and resources”: do you mean with both high and low economic and education resources?

Response: The comment is well acknowledged. The study had identified that ignorance was high among people with high economic, caste and education status. The sentence in the revised paper has been modified as below:

However, a qualitative formative study done in Nepal at community and health facility level has identified that ignorance about mental health issues was highly present among people of high economic and caste status and among people with more education and resources.

---

## [Decision Letter · Decision Letter 1]

7 Jan 2021

PONE-D-20-06276R1

Factors affecting utilization of mental health services from Primary Health Care (PHC) facilities of western hilly district of Nepal: The roles of stigma, awareness and availability

PLOS ONE

Dear Dr. Devkota,

Thank you for submitting your manuscript to PLOS ONE. After careful consideration, we feel that it has merit but does not fully meet PLOS ONE’s publication criteria as it currently stands. Therefore, we invite you to submit a revised version of the manuscript that addresses the points raised during the review process.

We look forward to receiving your revised manuscript.

Kind regards,

Pranil Man Singh Pradhan

Academic Editor

PLOS ONE

Reviewers' comments:

Reviewer's Responses to Questions

**Comments to the Author**

1. If the authors have adequately addressed your comments raised in a previous round of review and you feel that this manuscript is now acceptable for publication, you may indicate that here to bypass the “Comments to the Author” section, enter your conflict of interest statement in the “Confidential to Editor” section, and submit your "Accept" recommendation.

Reviewer #1: All comments have been addressed

Reviewer #2: All comments have been addressed

Reviewer #3: (No Response)

2. Is the manuscript technically sound, and do the data support the conclusions?

Reviewer #1: Yes

Reviewer #2: Partly

Reviewer #3: Yes

3. Has the statistical analysis been performed appropriately and rigorously? 

Reviewer #1: Yes

Reviewer #2: I Don't Know

Reviewer #3: N/A

4. Have the authors made all data underlying the findings in their manuscript fully available?

Reviewer #1: No

Reviewer #2: No

Reviewer #3: Yes

5. Is the manuscript presented in an intelligible fashion and written in standard English?

Reviewer #1: Yes

Reviewer #2: Yes

Reviewer #3: Yes

6. Review Comments to the Author

Reviewer #1: all comments have been addressed. Data is not available but authors have provided reasons for not making them available.

Reviewer #2: The authors have addressed the issues raised by the previous reviewer/s well.

Here are some of the major concerns regarding the study.

1. Title: The authors have explored various factors affecting service utilization in PHC facilities in the district rather than "the roles of stigma, awareness and availability". The roles of stigma, awareness and availability have not been explored in detail as well. So, the mention of "the roles of stigma, awareness and availability" in the title itself is not well justified.

2. How many of the primary health service providing centers in the study area had availability of mental health services? Non-availability of mental health service can be considered as a barrier to mental health service provision but definitely not a barrier to "utilization". This needs to be clarified further.

3. It is the lived experiences of those with mental illness and then their caretakers that provide more authentic information regarding individual and interpersonal factors, and even community level factors that facilitate or hinder service utilization. However, the narratives of those with mental illness and that of their caretakers have been grossly under-represented in the article. Would recommend to include more of the narratives from them.

4. Conclusion- Lines 678-683 (Community awareness can be provided through mass media, campaigns, and celebrations of days related to mental health. Mental illnesses support groups can be formed to reduce stigma at the community level and increase mental health literacy at the individual level through various programs. Inclusion of mental health education in the curriculum can also help arouse awareness and mental health literacy from an early age. These support groups and inclusion of mental health education in the curriculum can help reduce and/or eliminate stigma and increase awareness in community) have not been substantiated from the results or discussion portion of the manuscript. Better to either substantiate these claims in the conclusion or drop them out.

5. Calculation of "Inter-coder positive percent agreement"- It needs more elaboration. As per the table, all the codes from coder 2 must have matched with that of coder 1 for this gross calculation provided in the table. However, I would recommend more expert opinion in this regard.

Line 86- the inference of decreasing service utilization is not substantiated as the factors mentioned in the sentence refer to lack of service availability itself. So, this does not relate to "service utilization".

Line 97/98- "impairing family function(1), and increasing the risk of teenage childbearing(2) and domestic violence(3)"- this is cited from - 1. Keitner GI, Ryan CE, Miller IW, Kohn R, Bishop DS, Epstein NB. Role of the family in recovery and major depression. American Journal of Psychiatry 1995;152:1002-8. 2. Kessler RC, Berglund PA, Foster CL, Saunders WB, Stang PE, Walters EE. Social consequences of psychiatric disorders. II: Teenage parenthood. American Journal of Psychiatry 1997;154:1405-11. 3. Zlotnick C, Kohn R, Peterson J, Pearlstein T. Partner physical victimization in a national sample of American families: relationship to psychological functioning, psychosocial factors and gender. Journal of Interpersonal Violence 1998;13:156-65- in the article mentioned in your reference. So, while citing reference mentioned in other articles use appropriate referencing.

Line 199/200- psychiatrist being available nearby in order to manage any deviation from normal seen in the patients, if required.- Please clarify what does "manage any deviation from normal seen in the patients"

Line 244- mentally ill patients - patients with mental illness

Two of the themes(stigma, educational level) explained in the "Mental Illness and Help Seeking pathway" do not concern directly to help seeking pathways. Also, sudden mention of "help seeking pathway" does not seem relevant to factors affecting service utilization as mentioned in the title.

Line 402-404- It's obvious that one would not go to a facility where service is not available. So, this statement cannot be interpreted as a "barrier to service utilization". There would be no question of service utilization when service is not even available."Unavailability of service" is definitely a barrier to mental health service provision but not a barrier for "utilization". There would be no question of "service utilization" when the service does not even exist.

The verbatim given in lines 431-432 do not substantiate the claim of "commercialization of health services" as barrier to service utilization.

Similarly, referral from the PHCs themselves should not be interpreted as "barrier to service utilization" at PHC. (lines 428-435)

Lines 461-468- Here, issue related to stigma have been raised. But again, there are verbatim of service provider/administrator only. It would be more relevant if the issue of stigma has been raised by the service users.

Line 518-519- "whereas some traditional healers suggested not visiting health facilities" is not substantiated by the verbatim mentioned in line 522-23 "He told me to keep my523 wife there for 7 days in order to cure her, and said that doctors would take 15 days to cure her."

Lines 550-560- there's national mental health policy in Nepal (https://publichealthupdate.com/mental-health-policy-nepal/) and policy to provide mental health services from PHC services. Local bodies do not make their own mental health policies. So, the statements basically shows the lack of awareness among the participants. It's better to clarify these issues rather than just mentioning and interpreting the verbatim of participants.

Reviewer #3: This is an important area of research, thank you for the study. The paper contains rich information on barriers to use mental health service. However, I recommend following modifications:

Prevalence of Mental Disorders in Nepal may be added in the first paragraph after the global prevalence

59/60: advise to reframe the sentence “This is a stark reminder of the immense burden of the huge gap in mental health”

61: definition of treatment gap may not be needed; it has already been explained in lines 57-59

Treatment gap in nepal may also be added along with the treatment gap in India. The fact sheet on the finding of National MH survey of Nepal has been published.

82: what do you mean by senior staff?

110 – 113 aren’t these methodological issues relating to site selection?

151: it would be useful to state whether primary care centers (health posts/PHCCs etc) in the district provide mental health services as part of government’s program or NGO program. As availability is important factor for service use and also focus of this paper.

298-300: Suggest reviewing the write up. The line read as ‘educated people DO NOT have stigma at all’ which is not a consistent observation in researches published.

452: ‘Lack of psychiatrists, mental health experts or doctors in the PHC facilities acted as barriers for mental health service utilization’ I would suggest interpreting this as people interest to seek care from experts and not form general health care providers at PHCs. Otherwise, it suggests to place experts at PHCCs which is not the philosophy of primary care services.

The sections in settings (Methods) or the results did not provide information on whether the health care providers at the PHCCs are trained/supervised to provide MH service. This is relevant because without training and supervision, primary care providers generally will not be able to provide services in Nepal. Therefore, we would not be able to make inferences from statements where service users felt that PHCCs do not provide MH service. If PHCCs do not provide service for MH the problem is not in KNOWING, it is in AVAILABILITY. On the other hand, if health care providers are trained but people do not know it is different case. It would be great if authors could clarify on this.

633: suggest replacing ‘many mental health services’ to ‘basic mental health services’ … use psychotropic medications at all places

644: do you mean inadequate implementation? I am not convinced that every municipalities should have separate ‘Policy’ for mental health. What probably was meant by research participants is MH programs or even activities at municipal level.

678 – 685: These are the recommendations and do not seem to have come from the study participants as described in the results sections. I suggest to exclusively limit the conclusion on the finding from the study and not on the ‘opinion of the authors.

7. PLOS authors have the option to publish the peer review history of their article (what does this mean?). If published, this will include your full peer review and any attached files.

Reviewer #1: No

Reviewer #2: **Yes: **Madhur Basnet, MD(Psychiatry), Associate Professor, Department of Psychiatry, B. P. Koirala Institute of Health Sciences, Dharan, Nepal

Reviewer #3: No

---

## [Author Response · Author response to Decision Letter 1]

20 Feb 2021

Reviewer #1: all comments have been addressed. Data is not available but authors have provided reasons for not making them available.

Response: Respected Sir, the comment is well acknowledged.

Reviewer #2: The authors have addressed the issues raised by the previous reviewer/s well.

Here are some of the major concerns regarding the study.

1. Title: The authors have explored various factors affecting service utilization in PHC facilities in the district rather than "the roles of stigma, awareness and availability". The roles of stigma, awareness and availability have not been explored in detail as well. So, the mention of "the roles of stigma, awareness and availability" in the title itself is not well justified.

Response: Respected Sir, the comment is well acknowledged. The title has been refined as suggested.

2. How many of the primary health service providing centers in the study area had availability of mental health services? Non-availability of mental health service can be considered as a barrier to mental health service provision but definitely not a barrier to "utilization". This needs to be clarified further.

Response:Respected Sir, the comment is well acknowledged. The fact regarding presence of psychological counseling and referral services has been added.

3. It is the lived experiences of those with mental illness and then their caretakers that provide more authentic information regarding individual and interpersonal factors, and even community level factors that facilitate or hinder service utilization. However, the narratives of those with mental illness and that of their caretakers have been grossly under-represented in the article. Would recommend to include more of the narratives from them.

Response: Respected Sir, the comment is well acknowledged. Narratives from people with mental illnesses or their primary caretakers has been increased throughout the result section.

4. Conclusion- Lines 678-683 (Community awareness can be provided through mass media, campaigns, and celebrations of days related to mental health. Mental illnesses support groups can be formed to reduce stigma at the community level and increase mental health literacy at the individual level through various programs. Inclusion of mental health education in the curriculum can also help arouse awareness and mental health literacy from an early age. These support groups and inclusion of mental health education in the curriculum can help reduce and/or eliminate stigma and increase awareness in community) have not been substantiated from the results or discussion portion of the manuscript. Better to either substantiate these claims in the conclusion or drop them out.

Response: Respected Sir, the comment is well acknowledged. The necessary changes as suggested have been made in the conclusion section.

5. Calculation of "Inter-coder positive percent agreement"- It needs more elaboration. As per the table, all the codes from coder 2 must have matched with that of coder 1 for this gross calculation provided in the table. However, I would recommend more expert opinion in this regard.

Response:Respected Sir, the comment is well acknowledged. Expert opinion was provided by psychiatrist on the codes generated and the calculation of intercoder reliability. All the codes of Coder 2 did match with the codes of Coder 1. A simple percentage calculation was done then after.

Line 86- the inference of decreasing service utilization is not substantiated as the factors mentioned in the sentence refer to lack of service availability itself. So, this does not relate to "service utilization".

Response: Respected Sir, the comment is well acknowledged. The portion of the sentence “thereby decreasing service utilization” has been removed.

Line 97/98- "impairing family function(1), and increasing the risk of teenage childbearing(2) and domestic violence(3)"- this is cited from - 1. Keitner GI, Ryan CE, Miller IW, Kohn R, Bishop DS, Epstein NB. Role of the family in recovery and major depression. American Journal of Psychiatry 1995;152:1002-8. 2. Kessler RC, Berglund PA, Foster CL, Saunders WB, Stang PE, Walters EE. Social consequences of psychiatric disorders. II: Teenage parenthood. American Journal of Psychiatry 1997;154:1405-11. 3. Zlotnick C, Kohn R, Peterson J, Pearlstein T. Partner physical victimization in a national sample of American families: relationship to psychological functioning, psychosocial factors and gender. Journal of Interpersonal Violence 1998;13:156-65- in the article mentioned in your reference. So, while citing reference mentioned in other articles use appropriate referencing.

Response: Respected Sir, the comment is well acknowledged. Different studies that had identified the factors have been added as suggested.

Line 199/200- psychiatrist being available nearby in order to manage any deviation from normal seen in the patients, if required.- Please clarify what does "manage any deviation from normal seen in the patients"

Response: Respected Sir, the comment is well acknowledged. The sentence has been modified to provide understandable meaning.

Line 244- mentally ill patients - patients with mental illness

Response: Respected Sir, the comment is well acknowledged. The change as per suggestion has been made.

Two of the themes(stigma, educational level) explained in the "Mental Illness and Help Seeking pathway" do not concern directly to help seeking pathways. Also, sudden mention of "help seeking pathway" does not seem relevant to factors affecting service utilization as mentioned in the title.

Response: Respected Sir, the comment is well acknowledged. “Mental Illness and Help Seeking Pathway” was provided as background for the factors identified, which has been removed as suggested.

Line 402-404- It's obvious that one would not go to a facility where service is not available. So, this statement cannot be interpreted as a "barrier to service utilization". There would be no question of service utilization when service is not even available."Unavailability of service" is definitely a barrier to mental health service provision but not a barrier for "utilization". There would be no question of "service utilization" when the service does not even exist.

Response:Respected Sir, the comment is well acknowledged. The narration has been substituted with more relevant narrations.

The verbatim given in lines 431-432 do not substantiate the claim of "commercialization of health services" as barrier to service utilization.

Response: Respected Sir, the comment is well acknowledged. Presence of commercialization was acknowledged by elected representatives and the health administrators of the district.

Similarly, referral from the PHCs themselves should not be interpreted as "barrier to service utilization" at PHC. (lines 428-435)

Response: Respected Sir, the comment is well acknowledged. Substantial narrations have been provided.

Lines 461-468- Here, issue related to stigma have been raised. But again, there are verbatim of service provider/administrator only. It would be more relevant if the issue of stigma has been raised by the service users.

Response: Respected Sir, the comment is well acknowledged. Verbatim from service users has been added as suggested.

Line 518-519- "whereas some traditional healers suggested not visiting health facilities" is not substantiated by the verbatim mentioned in line 522-23 "He told me to keep my523 wife there for 7 days in order to cure her, and said that doctors would take 15 days to cure her."

Response: Respected Sir, the comment is well acknowledged. The necessary changes in the statement have been made.

Lines 550-560- there's national mental health policy in Nepal (https://publichealthupdate.com/mental-health-policy-nepal/) and policy to provide mental health services from PHC services. Local bodies do not make their own mental health policies. So, the statements basically shows the lack of awareness among the participants. It's better to clarify these issues rather than just mentioning and interpreting the verbatim of participants.

Response: Respected Sir, the comment is well acknowledged. Necessary changes as per suggestions have been made.

Reviewer #3: This is an important area of research, thank you for the study. The paper contains rich information on barriers to use mental health service. However, I recommend following modifications:

Prevalence of Mental Disorders in Nepal may be added in the first paragraph after the global prevalence

Response: Respected Madam/Sir, the comment is well acknowledged. Prevalence of Mental Disorders in Nepal has been added as suggested.

59/60: advise to reframe the sentence “This is a stark reminder of the immense burden of the huge gap in mental health”

Response: Respected Madam/Sir, the comment is well acknowledged. As the paragraph tried to present the burden of treatment gap in “inverted triangle” format, the sentence has been removed.

61: definition of treatment gap may not be needed; it has already been explained in lines 57-59

Treatment gap in nepal may also be added along with the treatment gap in India. The fact sheet on the finding of National MH survey of Nepal has been published.

Response: Respected Madam/Sir, the comment is well acknowledged. Definition of treatment gap has been removed and findings from National Mental Health Survey have been added as per suggestion.

82: what do you mean by senior staff?

Response: Respected Madam/Sir, the comment is well acknowledged. The term was used as such from the study cited. “Senior Community Medicine Assistant” as used in the study cited has been added.

110 – 113 aren’t these methodological issues relating to site selection?

Response: Respected Madam/Sir, the comment is well acknowledged. The sentence from the lines 110-113 have been kept in the “Setting” sub-heading of “Materials and Methods” as suggested. Moreover, the major mental disorder present in the diseases have also been kept within the sub-heading as it describes the status of mental disorder in the study area/site.

151: it would be useful to state whether primary care centers (health posts/PHCCs etc) in the district provide mental health services as part of government’s program or NGO program. As availability is important factor for service use and also focus of this paper.

Response: Respected Madam/Sir, the comment is well acknowledged. The availability of mental health services has been provided in the “Setting” sub-heading.

298-300: Suggest reviewing the write up. The line read as ‘educated people DO NOT have stigma at all’ which is not a consistent observation in researches published.

Response: Respected Madam/Sir, the comment is well acknowledged. The portion has been removed as it was used only to provide background in to the mental health context in the study site.

452: ‘Lack of psychiatrists, mental health experts or doctors in the PHC facilities acted as barriers for mental health service utilization’ I would suggest interpreting this as people interest to seek care from experts and not form general health care providers at PHCs. Otherwise, it suggests to place experts at PHCCs which is not the philosophy of primary care services.

Response: Respected Madam/Sir, the comment is well acknowledged. Changes in the sentence as per suggestions have been made.

The sections in settings (Methods) or the results did not provide information on whether the health care providers at the PHCCs are trained/supervised to provide MH service. This is relevant because without training and supervision, primary care providers generally will not be able to provide services in Nepal. Therefore, we would not be able to make inferences from statements where service users felt that PHCCs do not provide MH service. If PHCCs do not provide service for MH the problem is not in KNOWING, it is in AVAILABILITY. On the other hand, if health care providers are trained but people do not know it is different case. It would be great if authors could clarify on this.

Response: Respected Madam/Sir, the comment is well acknowledged. The availability of psychological counseling and referral services regardless of training and supervision provided has been added to the section.

633: suggest replacing ‘many mental health services’ to ‘basic mental health services’ … use psychotropic medications at all places

Response: Respected Madam/Sir, the comment is well acknowledged. The suggested change has been made.

644: do you mean inadequate implementation? I am not convinced that every municipalities should have separate ‘Policy’ for mental health. What probably was meant by research participants is MH programs or even activities at municipal level.

Response: Respected Madam/Sir, the comment is well acknowledged. The changes as per suggestions has been made.

678 – 685: These are the recommendations and do not seem to have come from the study participants as described in the results sections. I suggest to exclusively limit the conclusion on the finding from the study and not on the ‘opinion of the authors.

Response: Respected Madam/Sir, the comment is well acknowledged. The conclusion has been modified as per suggested.

---

## [Decision Letter · Decision Letter 2]

13 Apr 2021

Factors affecting utilization of mental health services from Primary Health Care (PHC) facilities of western hilly district of Nepal

PONE-D-20-06276R2

Dear Dr. Devkota,

We’re pleased to inform you that your manuscript has been judged scientifically suitable for publication and will be formally accepted for publication once it meets all outstanding technical requirements.

Kind regards,

Pranil Man Singh Pradhan

Academic Editor

PLOS ONE

Additional Editor Comments (optional):

Reviewers' comments:

Reviewer's Responses to Questions

**Comments to the Author**

1. If the authors have adequately addressed your comments raised in a previous round of review and you feel that this manuscript is now acceptable for publication, you may indicate that here to bypass the “Comments to the Author” section, enter your conflict of interest statement in the “Confidential to Editor” section, and submit your "Accept" recommendation.

Reviewer #3: All comments have been addressed

2. Is the manuscript technically sound, and do the data support the conclusions?

Reviewer #3: Yes

3. Has the statistical analysis been performed appropriately and rigorously? 

Reviewer #3: Yes

4. Have the authors made all data underlying the findings in their manuscript fully available?

Reviewer #3: Yes

5. Is the manuscript presented in an intelligible fashion and written in standard English?

Reviewer #3: Yes

6. Review Comments to the Author

Reviewer #3: Thank you for addressing the comments. The second paragraph (64-72) under introduction section may be revised to make it concise.

The rest seem ok to me.

7. PLOS authors have the option to publish the peer review history of their article (what does this mean?). If published, this will include your full peer review and any attached files.

Reviewer #3: No

---

## [Editor Report · Acceptance letter]

23 Apr 2021

PONE-D-20-06276R2 

Factors affecting utilization of mental health services from Primary Health Care (PHC) facilities of western hilly district of Nepal 

Dear Dr. Devkota:

I'm pleased to inform you that your manuscript has been deemed suitable for publication in PLOS ONE. Congratulations! Your manuscript is now with our production department. 

Kind regards, 

on behalf of

Dr. Pranil Man Singh Pradhan 

Academic Editor

PLOS ONE